DATA RELEASE

# The first complete mitochondrial genome of *Diadema antillarum* (Diadematoida, Diadematidae)

Audrey J. Majeske[1,2,*], Alejandro J. Mercado Capote[1], Aleksey Komissarov[3], Anna Bogdanova[3], Nikolaos V. Schizas[4], Stephanie O. Castro Márquez[1,2], Kenneth Hilkert[2], Walter Wolfsberger[1,2,5] and Tarás K. Oleksyk[1,2,5]

1 University of Puerto Rico at Mayagüez, Department of Biology, Call Box 9000, Mayagüez, PR 00681, USA
2 Oakland University, Department of Biological Sciences, 118 Library Drive, Rochester, MI 48309, USA
3 Applied Genomics Laboratory, SCAMT Institute, ITMO University, Lomonosova 9 Str., Saint Petersburg 197101, Russia
4 University of Puerto Rico at Mayagüez, Department of Marine Sciences, Call Box 9000, Mayagüez, PR 00681, USA
5 Uzhhorod National University, Faculty of Biology, Universytets'ka St, 14, Uzhhorod, Zakarpattia Oblast 88000, Ukraine

## ABSTRACT

The mitochondrial genome of the long-spined black sea urchin, *Diadema antillarum*, was sequenced using Illumina next-generation sequencing technology. The complete mitogenome is 15,708 bp in length, containing two rRNA, 22 tRNA and 13 protein-coding genes, plus a noncoding control region of 133 bp. The nucleotide composition is 18.37% G, 23.79% C, 26.84% A and 30.99% T. The A + T bias is 57.84%. Phylogenetic analysis based on 12 complete mitochondrial genomes of sea urchins, including four species of the family Diadematidae, supported familial monophyly; however, the two *Diadema* species, *D. antillarum* and *D. setosum* were not recovered as sister taxa.

**Subjects** Animal and Plant Sciences, Genetics, Marine Biology

**Submitted:** 27 May 2022

* Corresponding author. E-mail: amajeske@oakland.edu

Preprint submitted at https://doi.org/10.1101/2022.10.05.510842

## DATA DESCRIPTION

The long-spined black sea urchin, *Diadema antillarum* (Diadematoida, Diadematidae; NCBI:txid105358; urn:lsid:marinespecies.org:taxname:124332), is a marine benthic invertebrate inhabiting the shallow waters of the western Atlantic Ocean and Caribbean Sea. It is an important herbivore and keystone species that helps maintain healthy coral reef systems along its coastal marine habitats [1–9]. After the human impact of overharvesting larger herbivorous fishes and the disappearance of larger vertebrates, *D. antillarum* was plentiful. Along with other smaller herbivore fishes, they served as the primary grazers maintaining the health of a coral-dominated reef system, up until the mid-1980s [9–11]. After the historic 1983/84 die-off event of this species, which was presumably caused by an unknown water-borne pathogen, repeated die-off events occurred in the 1990s as well as in the current year [12]. These events have collectively decimated, as well as wiped out, populations in some localities across the Caribbean [13–15]. Monitoring and recovery efforts have been promising, but future die-off events will likely occur as global climate change continues [16–18].

## CONTEXT

Until now, only a few genes have been described in the mitochondrial genome of *D. antillarum*, including partial sequences of *COI*, *COII* and *ATP6*, as well as assembled sequences of *ATP8* and *tRNA^{Lys}* [19, 20]. While over 40 mitochondrial genomes of different sea urchin species have been completed so far, this includes only three species within the family Diadematidae, which contains 12 genera. Of these, both species of the genus *Echinothrix* have complete mitochondrial genomes in addition to one of the seven species within the genus *Diadema*, specifically *D. setosum*. Here, we report the complete mitochondrial genome of *D. antillarum* and we explore the utility of whole mitochondrial genomes to place *D. antillarum* phylogenetically in the Diadematidae tree.

## METHODS

### Sample collection and DNA extraction

An adult sea urchin was collected in Puerto Rico (18˚20′35.2″N 67˚15′40.5″W). A sample containing whole coelomocytes was withdrawn from the animal through the Aristotle's Lantern using a sterile 23-gauge needle connected to a 5-mL syringe. The animal was photographed and then returned to its habitat at the collection site; it was not retained as a voucher specimen for this study. The sample was held on ice during transport to the University of Puerto Rico Mayagüez (UPRM). Total DNA was extracted from coelomocytes per company instructions using an a DNeasy® Blood and Tissue Kit (Qiagen, Inc.). The sample quality and concentration were assessed using a NanoDrop 2000 spectrophotometer (ThermoFisher Scientific) prior to shipment for sequencing at the Psomagen (Macrogen USA) laboratory in Rockville, MD, USA. Approval for the sample collection was obtained from the Department of Natural and Environmental Resources of Puerto Rico (O-VS-PVS15-AG-00046-01082018).

### Sequencing and bioinformatics analysis

To prepare the DNA sample for sequencing, a library of sequences was generated using a TruSeq® DNA PCR Free (350) library preparation kit (Illumina, Inc.). Quality and quantitation verification of libraries was performed using a Bioanalyzer. Next, an Illumina HiSeq 2500 platform generated paired-end 151-base pair (bp) sequence reads. The sequencer produced a total of 60.8 Gbp and 402,874,618 paired-end reads, of which 88.21% had a quality score of ≥Q30.

Raw read candidates for the mitochondrial genome were extracted bioinformatically from the total Illumina data representing both nuclear and mitochondrial genomes using Cookiecutter2 software [21, 22]. The reads were trimmed using a custom program called v2trim [22].

Next, the list of *k*-mers was formed in two stages. First, we used the best-known reference genome of sea urchins (*Strongylocentrotus purpuratus*, NC_001453.1 [23]) to assemble the draft mtDNA contigs of *D. antillarum* using the SPAdes genome assembler v3.15.4 (RRID:SCR_000131) [24]. Assembled contigs were then searched using the National Center for Biotechnology Information's BLAST web interface [25] to find the closest reference sequence matching the species *Echinothrix diadema*, with the accession number KX385836.1. In the second stage, we constructed the *k*-mer list using this closest reference sequence. Extracted reads were then assembled with SPAdes using default parameters. To verify assembly quality, the extracted reads were mapped back to the full-length mtDNA



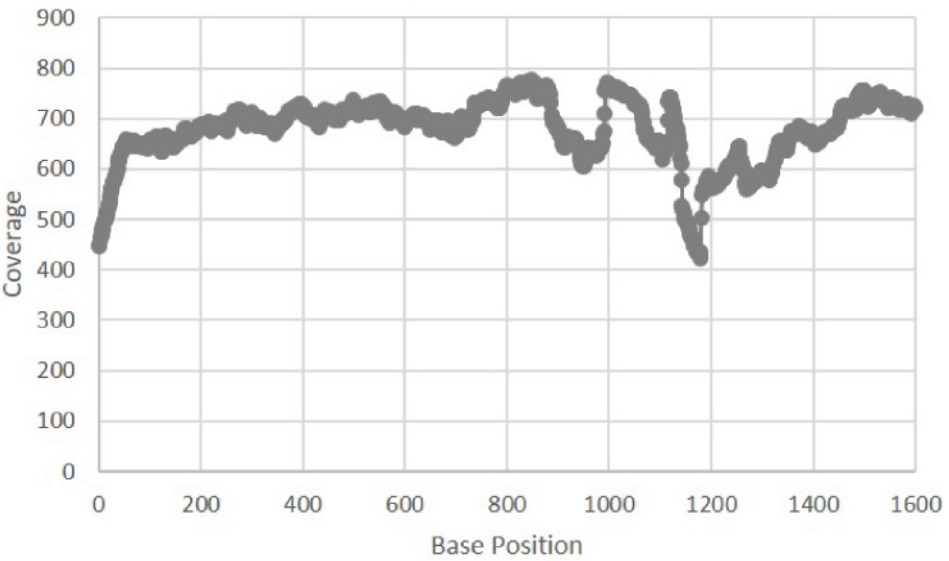

**Figure 1.** **Mitogenome assembly coverage map of *Diadema antillarum*.**
A per-base coverage data table was generated using Unipro UGENE. This graph was generated in Microsoft Excel (2019). The *X*-axis shows the base position of the mitogenome sequence. The *Y*-axis shows the assembly coverage at that base. The length of the mitogenome is 15,708 bp.

assembly with BWA-MEM2 (RRID:SCR_022192) using the default parameters [26, 27]. The coverage plot was computed with bedtools genomecov with −d and −split parameters (BEDTools RRID:SCR_006646) [28]. In addition, a per-base coverage data table was generated using the NGS data analysis tools provided in the Unipro UGENE software program [29]. A graph of this data table was generated in Microsoft Excel (2019) and is presented as Figure 1. Finally, the assembly start was rotated to tRNA-Phe. Following assembly, an online annotation was performed using the MITOS web server [30] with following manual verification of each predicted RNA and protein. The results from this annotation were used to generate a mitochondrial map in Geneious Prime v2022.1.1 (Figure 2) [31]. The extracted reads, bam coverage files and the reproducible commands list are available in our GitHub repository [32].

## Phylogenetic analysis
Alignment and phylogenetic analysis were constructed in MEGA v11.0.11 [33]. Analysis included the complete mitochondrial genomes of 12 representative species from the orders Camarodonta (families Temnopleuridae, Echinometridae and Parechinidae), Arbacioida (family Arbaciidae), Temnopleuroida (family Toxopneustidae), Cidaroida (family Cidaridae), Echinoida (family Strongylocentrotidae), and Diadematoida (family Diadematidae) plus the sea cucumber, *Apostichopus japonicus*. A MUSCLE alignment method was implemented using the default parameters in the MEGA11 program. Another sequence alignment editor, BioEdit, was used to generate a sequence identity matrix of the aligned sequences [34]. We performed IQ-TREE analysis to determine the best-fit model of substitution among the mitogenome sequences [35]. A maximum likelihood (ML) tree was generated with the mitogenome nucleotide DNA sequences of the 12 sea urchin species and the sea cucumber included as an outgroup. For phylogenetic analysis, we chose a comprehensive model of



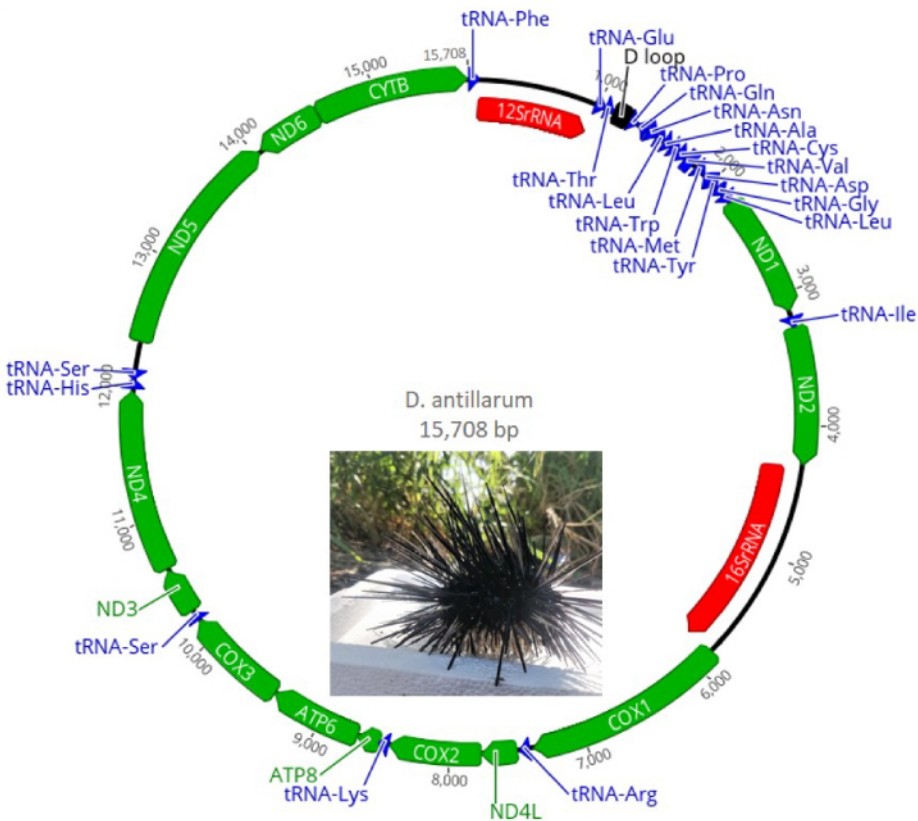

**Figure 2. Mitochondrial genome map of *Diadema antillarum*.**
Arrow shapes are colored to indicate tRNA genes (blue), rRNA genes (red) and protein-coding genes (green). The black rectangular shape refers to the noncoding control region or D loop. The directions of the arrows indicate direction of transcription on the H-strand (arrow points to the right) and L-strand (arrow points to the left). The map was generated in Geneious Prime v2022.1.1. The animal image is the specimen used for sampling in this study. Image taken by Alejandro Mercado Capote.

parameters that resulted in branches containing the best supported bootstrap values (out of 500 bootstrap iterations) for the resulting consensus tree. The ML tree was generated using a Tamura–Nei model of evolution [36]. A discrete Gamma distribution was used to model evolutionary rate differences among sites (five categories (+*G*, parameter = 0.3129)). Initial tree(s) for the heuristic search were obtained automatically by applying Neighbor-Join and BioNJ algorithms to a matrix of pairwise distances estimated using the Tamura–Nei model, and then the ML tree was generated by selecting the topology with superior log likelihood value. Additional sequence alignments and phylogenetic analysis was performed on available sequences for *Echinothrix* spp. and *Diadema* spp. for each of the three genes: 16S ribosomal RNA (partial sequence), ATPase genes (including ATP synthase subunit 6 gene partial coding sequence and tRNA-Lys gene partial sequence ATP8 gene complete coding sequence and ATP6 gene partial coding sequence) and cytochrome oxidase subunit 1 (CO1) (partial coding sequence). For these additional analyses, a MUSCLE alignment was generated using the default parameters in MEGA11. In addition, we extracted the portions of the mitogenome for *E. diadema* (KX385836), *E. calamaris* (MK609484), *D. setosum* (KX385835) and *D. antillarum* (present study) that corresponded to each gene, by generating an initial alignment using available sequences from the same genera for each of the three



genes. A ML tree was obtained for each gene using a bootstrapped (500 iterations) general time reversible (GTR) model with invariant (I) substitution rates among the nucleotides. The initial trees were obtained by applying Neighbor-Join and BioNJ algorithms to a matrix of pairwise distances estimated using the Tamura–Nei model, and then the ML tree was generated by selecting the topology with superior log likelihood value.

## DATA VALIDATION AND QUALITY CONTROL

### Signatures of the mitogenome

The circular mitogenome of *D. antillarum* is 15,708 bp in length, comprising two rRNA genes, 22 tRNA genes, a noncoding control region, and 13 protein-coding genes that are common in other echinoderms, as well as the order of the genes [37–39]. Most of the genes are encoded on the H-strand, except for one protein-coding gene, *ND6*, and five tRNA genes, including $tRNA^{Gln}$, $tRNA^{Ala}$, $tRNA^{Val}$, $tRNA^{Asp}$, and $tRNA^{Ser}$, which are encoded on the L-strand [40]. Most of the protein-coding genes use the start codon ATG, with the exception of *ATP8*, which starts with GTG. The length of the rRNA genes is 896 bp for *12S rRNA* and 1555 bp for*16S rRNA*. The control region is 133 bp in length and contains the typical G repeat that is found in other echinoderms. This noncoding region is located at base positions 1111 to 1243, and is positioned between the genes $tRNA^{Thr}$ and $tRNA^{Pro}$.

The composition of the nucleotides for the mitogenome of *D. antillarum* was calculated in MEGA11 as 18.37% G, 23.79% C, 26.84% A and 30.99% T. The A + T bias was also calculated in MEGA, which is 57.84% for *D. antillarum* – slightly lower, but comparable, with that of *D. setosum*, which is 58.19%. Compared with the two other *Echinothrix* species in the same family as *D. antillarum*, the A + T bias in the mitogenome of *D. antillarum* is slightly higher than that of *E. diadema* (57.61%) and *E. calamaris* (56.43%). Nonetheless, when compared with species in different orders, the mitochondrial A + T bias of *D. antillarum* is usually lower than other species in the following orders: Cidaroida (*Stylocidaris reini*, 59.93%; *Prionocidaris baculosa*, 59.14%; *Eucidaris tribuloides*, 59.70%), Camarodonta (*Echinometra mathaei*, 59.21%; *Heterocentrotus mamillatus*, 58.91%; *Heliocidaris crassispina*, 58.89%), and Echinoida (*Strongylocentrotus droebachiensis*, 58.96%; *S. intermedius*, 58.92%; *S. purpuratus*, 58.98%).

### Phylogenetic tree

The bootstrap consensus ML tree is shown in Figure 3. This was inferred from 500 replicates through clustering of the associated taxa [41]. The results of the IQ-TREE analysis indicated that the best model of substitution was the most comprehensive model tested, the GTR+F+R2 model (general time reversible model with unequal rates and unequal base frequency with FreeRate rate heterogeneity model) [42]. The results of this test included a ML tree lacking a bootstrap analysis, which is available in GitHub [32]. In addition, a ML tree was generated in MEGA11 program using the suggested GTR model of evolution with invariant (or unequal) substitution rates among sites. The two resulting ML consensus trees from MEGA11, plus the ML tree generated by IQ-TREE analysis, had identical topologies with regards to the species within the family Diadematidae. Of the two bootstrapped ML trees produced in MEGA, the tree presented in this study tended to show higher bootstrap values for all branches within the tree.

The tree topology shown in this study indicated that *D. antillarum* was more closely related to *E. diadema* than to *D. setosum*. The next most closely related taxon to these three

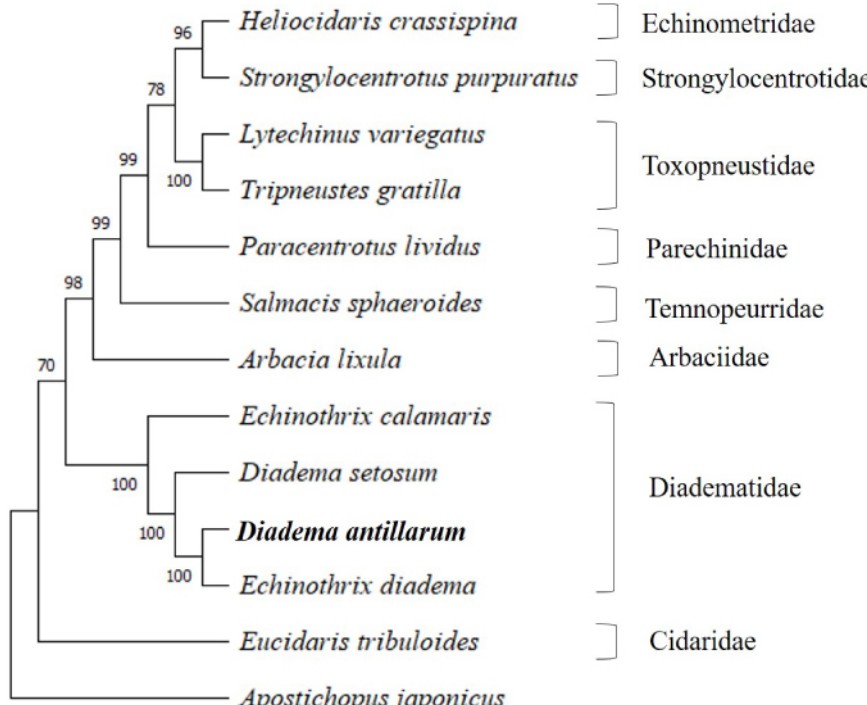

**Figure 3. Phylogenetic analysis of mitochondrial genomes indicates that *Diadema antillarum* is more closely related to *Echinothrix diadema* than to *D. setosum*.**
A maximum likelihood consensus tree of sequences representing 12 sea urchin species and a sea cucumber (outgroup). Genbank accession numbers for taxa on the tree include: *H. crassispina* (KC479025.1 [43]), *S. purpuratus* (NC_001453.1 [23, 44, 45]), *L. variegatus* (NC_037785.1 [37]), *T. gratilla* (KY268294.1 [46]), *P. lividus* (J04815.1 [47]), *S. sphaeroides* (KU302103.1 [41]), *A. lixula* (X80396.1 [39]), *E. calamaris* (NC_050274.1 [48]), *D. setosum* (KX385835.1 [49]), *E. diadema* (KX385836.1 [50]), *E. tribuloides* (MH614962.1 [51]) and *A. japonicus* (NC_012616 [52]). Branch lengths are equalized and do not include evolutionary divergence times. Bootstrap support values are included on the tree branches.

was *E. calamaris*. For *E. diadema*, *E. calamaris* and *D. setosum*, the length of these mitochondrial genomes are 15,712 bp, 15,716 bp and 15,708 bp, respectively. A previous study reporting the complete mitogenome of *D. setosum* included a ML phylogenetic tree with a highly supported sister clade (bootstrap value of 100) containing two taxa: *E. diadema* and *D. setosum* [49]. Furthermore, the results of our sequence identity matrix of mitogenomes between *D. antillarum* and *E. diadema* was 96.7% identical, whereas, *D. antillarum* vs. *D. setosum* or *D. antillarum* vs. *E. calamaris* had 86.3% and 80% identity, respectively. Additionally, the mitogenomes between *E. diadema* and *D. setosum* was 86.2% identical in sequence. This suggests that the overall higher similarity between *D. antillarum* and *E. diadema*, as opposed to *D. setosum*, could be associated with the higher A + T bias in the mitogenome. However, the mitogenome sequences of *E. diadema* and *D. setosum* were generated from specimens collected in the South China Sea from the same research group [49, 50]. Given there are other *Diadema* species in the South China Sea that are morphologically similar to *E. diadema* (including *D. setosum* and *D. savignyi*), the reported mitogenome of *E. diadema* may have been sampled from a *Diadema* species, namely *D. savignyi*. Unfortunately, the mitogenome for *D. savignyi* has not been completed or is otherwise unavailable. Nonetheless, to provide evidence for this claim, we performed sequence analysis for three additional genes (*16S*, *ATPase* and *CO1*) for available data from



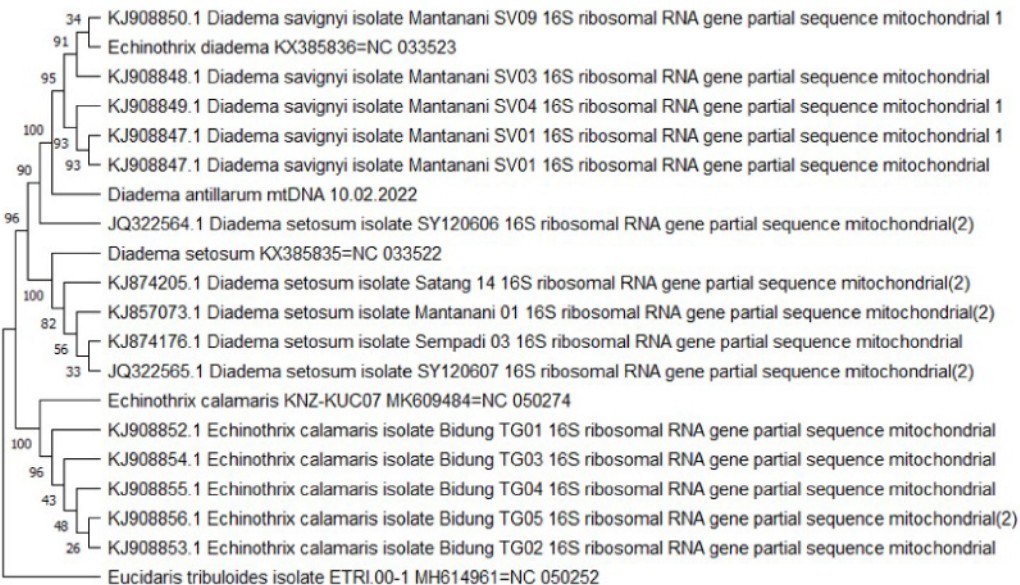

**Figure 4.** Phylogenetic analysis of 16S ribosomal RNA sequences for *Diadema and Echinothrix* spp.
A maximum likelihood bootstrapped consensus tree of 19 partial gene sequences for *Diadema and Echinothrix* spp. plus *Eucidaris tribuloides* as an outgroup. A section of the mitogenome for *E. diadema* (KX385836), *D. setosum* (KX385835), *E. calamaris* (MK609484), *D. antillarum* (10.02.2022) and *E. tribuloides* (ETRI 00-1) that corresponded to the 16S gene was extracted by generating an initial alignment. To do this, the *E. calamaris* 16S gene-specific sequences were aligned to the entire *E. tribuloides, E. calamaris* and *E. diadema* mitogenomes. Then, the *D. setosum and D. savignyi* gene-specific sequences for 16S were aligned to the entire *E. tribuloides* and *D. antillarum* mitogenomes. Lastly, the *D. setosum* gene-specific sequences listed in the tree were aligned to the entire *E. tribuloides* and *D. setosum* mitogenomes. The rest of the mitogenome for each species was trimmed to only reflect the portion pertaining to the 16S gene sequence. Thus, the taxa on the tree named Diadema antillarum mtDNA 10.02.2022, Echinothrix diadema KX385836=NC 033523, Diadema setosum KX38535=NC 033522, Echinothris calamaris KNZ-KUC07 MK609484=NC 050274 and Eucidaris tribuloides isolate ETRI 00-1 MH614961=NC 050252 include only the 16S gene sequences. Branch lengths are equalized and do not include evolutionary divergence times. Bootstrap support values are included on the tree branches.

species of the genera *Echinothrix* and *Diadema*. Results of the tree topologies for all three genes indicated that the specific gene sequences extracted from the mitogenome of *E. diadema* are more closely related to the particular gene sequence obtained from *Diadema* sp. rather than those from *Echinothrix* sp. (Figures 4, 5, and 6). In particular, the specific gene sequences extracted from the mitogenome of *E. diadema* were placed as a sister clade with *D. savignyi* (see Figures 4–6), which was placed next to a larger group of clades including *D. africanum* (see Figure 4), *D. antillarum,* (see Figures 4–6), *D. mexicanum* (see Figure 4) and *D. setosum* (see Figures 4–6). This was more distantly related to groupings of clades that included *E. diadema* (see Figure 4) and *E. calamaris* (see Figures 4–6).

Furthermore, it is important to note that the habitat ranges of both *E. diadema* and *D. setosum* are in the Indo-Pacific region, which is distinct from the habitat range of *D. antillarum* typically found in the western Atlantic Ocean, the Caribbean Sea, the tropical coasts of South America down to Brazil, and from Bermuda to Florida. A new subspecies of *D. antillarum*, called *D. antillarum ascensionis*, has been identified in the eastern Atlantic, which is similar to the mid-Atlantic species of *D. antillarum* [53]. This new subspecies is also similar to *D. africanum* found in the eastern Atlantic islands, from the Madeira islands to the Guinean Gulf, including the Salvage, Canary, Cape Verde [54], and Sâo Tome islands [19]. While *D. antillarum* has been repopulating areas, which has extended their habitat range,

*A. J. Majeske et al.*

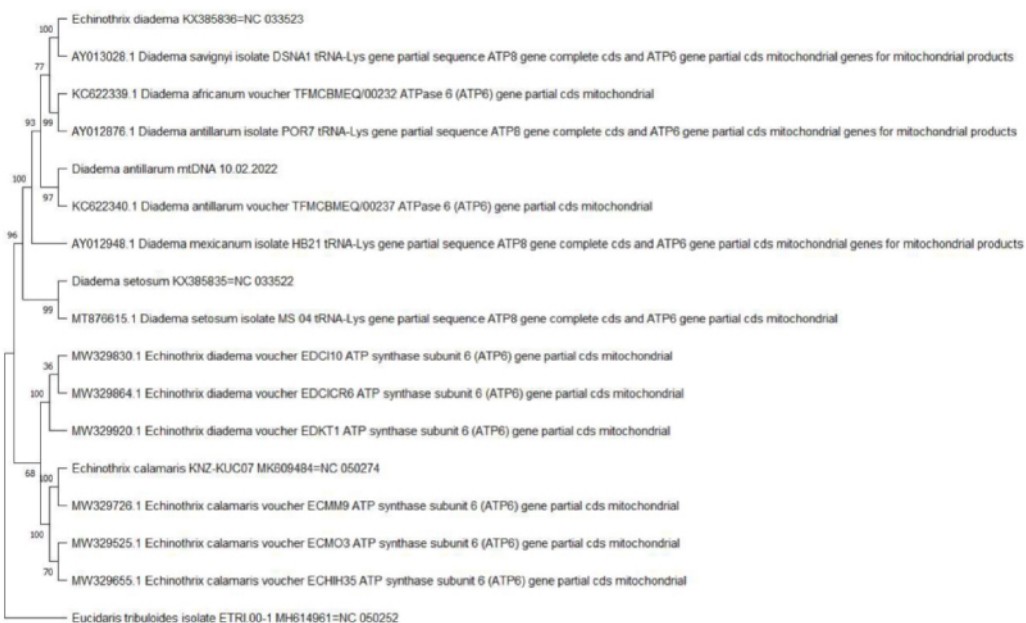

**Figure 5.** **Phylogenetic analysis of ATPase genes for *Diadema and Echinothrix* spp.**
ATP synthase subunit 6 gene (partial coding sequence) and ATP synthase subunit 8 gene (complete coding sequence) were included in the analysis, as well as tRNA-Lys gene (partial sequence), which was included in some of the taxa. A maximum likelihood bootstrapped consensus tree is shown for 16 sequences of *Diadema and Echinothrix* spp. plus *Eucidaris tribuloides* as an outgroup. A section of the mitogenome for *E. diadema* (KX385836), *D. setosum* (KX385835), *E. calamaris* (KNZ-KUC07), *D. antillarum* (10.02.2022) and *E. tribuloides* (ETRI 00-1) that corresponded to the specified sequence was extracted by generating an initial alignment. To do this, the *E. calamaris* and *E. diadema* gene-specific sequences listed in the tree were aligned to the entire *E. tribuloides, E. diadema* and *E. calamaris* mitogenomes. Then, the *D. antillarum, D. africanum, D. setosum, D. savignyi* and *D. mexicanum* gene-specific sequences listed in the tree were aligned to the entire *E. tribuloides* and *D. antillarum* mitogenomes. Lastly, the *D. setosum* gene-specific sequences listed in the tree were aligned to the entire *E. tribuloides* and *D. setosum* mitogenome. The rest of the mitogenome for each species was trimmed to only reflect the portion pertaining to the ATPase gene sequences. Thus, the taxa on the tree named *Diadema antillarum* mtDNA 10.02.2022, *Echinothrix diadema* KX385836=NC 033523, *Diadema setosum* KX38535=NC 033522, *Echinothrix calamaris* KNZ-KUC07 MK609484=NC050274 and *Eucidaris tribuloides* isolate ETRI 00-1 MH614961=NC 050252 include only the ATPase gene sequences. Branch lengths are equalized and do not include evolutionary divergence times. Bootstrap support values are included on the tree branches.

we are not aware of any *D. africanum* migration as far west as Puerto Rico, in the Caribbean Sea. In addition, when identifying the specimen prior to collection for this study, it lacked the iridophores, typical of *D. africanum*, which are visible in sunlight. While the *ATPase* sequence analysis presented in this study does reflect that there are similarities in these sequences between *D. antillarum*, *D. africanum* and *D. mexicanum*, the *ATPase* gene sequence that was extracted from the mitogenome used in this analysis most closely resembles that of another *D. antillarum* (see Figure 4). Thus, we believe that the species used in this study was correctly identified. Yet, as additional mitogenome sequences become available, the phylogenetic tree describing the relationships between genera within Diadematidae should become more thorough.

## REUSE POTENTIAL

These results may provide some insight into the dispersal and speciation events of the family Diadematidae. However, more data are needed from other genera to draw further conclusions about the evolution and adaptation of this family lineage. The mitogenome sequence produced here can serve as a reference sequence for this species of sea urchin. In

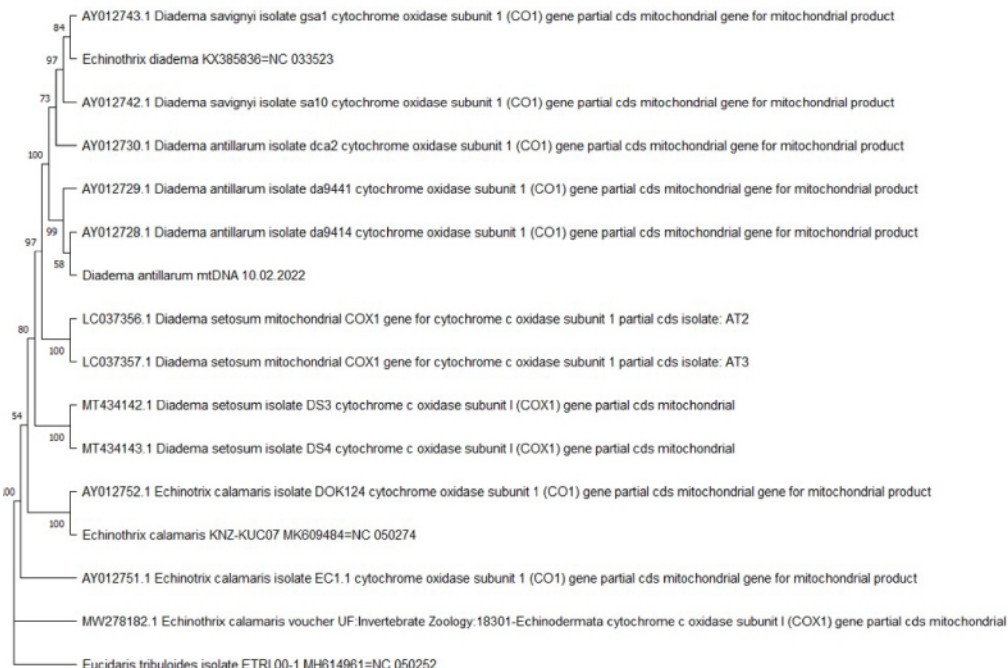

**Figure 6.** **Phylogenetic analysis of cytochrome oxidase 1 (*CO1*) genes for *Diadema and Echinothrix* spp.** A maximum likelihood bootstrapped consensus tree is shown for 15 partial coding sequences of *Diadema and Echinothrix* spp. plus *Eucidaris tribuloides* as an outgroup. A section of the mitogenome for *E. diadema* (KX385836), *E. calamaris* (KNZ-KUC07), *D. antillarum* (10.02.2022) and *E. tribuloides* (ETRI 00-1) that corresponded to the *CO1* gene was extracted by generating an initial alignment. To do this, the *E. calamaris* gene-specific sequences listed in the tree were aligned to the entire *E. tribuloides*, *E. diadema* and *E. calamaris* mitogenome. Separately, the gene-specific sequences listed in the tree from *D. savignyi*, *D. antillarum*, and *D. setosum* were aligned to the entire *E. tribuloides* and *D. antillarum* mitogenome. The rest of the mitogenome for each species was trimmed to only reflect the portion pertaining to the *CO1* gene sequence. Thus, the taxa on the tree named *Diadema antillarum* mtDNA 10.02.2022, *Echinothrix calamaris* KNZ-KUC07 MK609484=NC050274, *Echinothrix diadema* KX385836=NC 033523 and *Eucidaris tribuloides* isolate ETRI 00-1 MH614961=NC 050252 include only the *CO1* gene sequences. Branch lengths are equalized and do not include evolutionary divergence times. Bootstrap support values are included on the tree branches.

addition, the nuclear sequences generated here will be included in a larger study to assemble the whole genome for this species. Given the necessity of *D. antillarum* for maintaining the health and current structure of the remaining coral-dominated coastlines in the face of global climate change, it will be important to include the genetics of this species in future conservation and population genetics studies.

## DATA AVAILABILITY

The mitochondrial genome sequence has been deposited at DDBJ/ENA/GenBank under the accession number ON725136. The data is linked to the NCBI BioProject and BioSample numbers PRJNA839760 and SAMN28553754, respectively. All intermediate files are available in the GigaDB repository [55].

## DECLARATIONS

## List of abbreviations

bp: base pair; Gbp: gigabase pairs; GTR: general time reversible; ML: maximum likelihood.

## Ethical approval

Approval for the collection of sea urchin samples was obtained from the Department of Natural and Environmental Resources of Puerto Rico (O–VS-PVS15-AG-00046-01082018).

## Consent for publication

Not applicable.

## Competing Interests

The authors declare that they have no competing interests.

## Funding

Funding for the project was provided by start-up funds granted to Tarás Oleksyk by Oakland University.

## Acknowledgements

The authors would like to thank Heidi D. Morales Díaz for assistance with the animal sample collection.

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
