## [Reviewer Report]

Comments on revised manuscriptThe revised manuscript of Majeske et al is much improved in comparison to the initial submission. Some of the questions raised in the previous review, however, remain open and other new aspects have appeared.

Open issues:
2) Please provide the repository number and institution where the voucher specimen has been deposited
--> this issues has not been addressed in the revised version; it is unclear if a voucher specimen has been deposited or not, where it is stored and which inventory number it has; if the specimen has not been retained, this is unfortunate, but not a huge issue - it still needs to be clearly/openly stated

3) Did you verify the identification and made sure that this is D. antillarum rather than D. africanum (which allegedly has repopulated some D. antillarum habitats in the Caribbbean and GoM) 
--> this issue too has not been addressed; at the very least I would expect a statement that the authors were aware of this second Atlantic Diadema species and how they made sure they really had D. antillarum

7) Please provide a coverage graph
--> the coverage graph is mentioned in the text, but not provided in the paper

9) Please explain what exactly was used for the analysis - the full nucleotid sequence including non-coding regions, just the CDS of the protein coding genes, or ?
--> this is still unclearly formulated in the paper - I assume the whole mitogenome sequence was used, but the wording is very ambiguous; this needs to be very clearly stated in the material and metods section

11) Please explain the choice of the model used in the analysis - was some Modeltest run?
--> this information is still lacking


New issues:

A) The description of the assembly process is still rather unclear - this needs to be better explained. For example, was any kind of preprocessing (read triming etc.) done? Which parameters were chosen for the various programms employed? How did the two-stage read extraction process really work - the wording in the manuscript is very unclear regarding this aspect

B) The raw data need to be deposited in the GenBank Short Read Archive (SRA), in the Github repository only the extracted mitochondrial reads are available - this is insufficient to repeat the assembly process and analyses carried out in the present manuscript

C) The fasta file included in the Github repository has 23 positions that are redundant (overlapping with the start of the sequence) - they need to be removed before submisson

D) There is some inconsistence on the length of the mitogenome, the text says 15,708, the figure says 15,707 - the latter, judging form the files in your Github repository, is correct --> please make sure the information given is consistent

E) No information is given on the reason for chosing the particular evolutionary model that has been used in the phylogenetic analysis

F) The phylogenetic analysis has been done by NJ-methods, which are fast but can subject to a lot of problems, it would be better to use MAximu Likelihood (or Bayesian) methods

G) The authors have made an important discovery in relation to the mitogenome deposited as "Echinothrix diadema" in GenBank. Rather than to speculate on the reasons that is the sister of D. antillarum in their analysis the authors should simply which of their hypotheses (AT-bias vs. misidentification) is correct. All the tools that are needed are already available in Genbank! There is an extensive dataset of three mitochondrial markers (12S, ATP6, ATP8; https://www.ncbi.nlm.nih.gov/popset/?term=MW329515 etc.) available for Echinothrix, which includes hundreds of sequences and encompases material from the complete geographical range of the genus (Coppard et al. 2021 https://www.nature.com/articles/s41598-021-95872-0). In addition, there are 16S sequences available for D. savignyi, the suspected candidate of the misidentification. I have downloaded these sequences and run preliminary analyses with with a subset of the sequences. These clearly show that the "E. diadema" mitogenome has nothing to do with true E. diadema and that it is a Diadema. While the data basis for Diadema is less extensive than for Echinothrix there are 16S sequences of D. savignyi (GenBank PopSet: 673458050) that are identical to part of the 16S sequence of the alledged "E. diadema" mitogenome. Thus I am convinced that the second hypothesis (misidentification) of the authors is correct. This is an important finding that should be discussed in depth in the manuscript. I am includid the alignments and trees that I made in the attachment - similar analyses and trees should be included in the manuscript.

Link to download the attachments: https://we.tl/t-y7ypbnZYPQ

Summing up, I recommend acceptance after major revision.
Kind regards
Andreas Kroh, NHM Vienna, 11/9/2022

---

## [Reviewer Report]

Reviewer name and names of any other individual's who aided in reviewer Jose LopezDo you understand and agree to our policy of having open and named reviews, and having your review included with the published papers. (If no, please inform the editor that you cannot review this manuscript.)YesIs the language of sufficient quality?YesPlease add additional comments on language quality to clarify if needed
Are all data available and do they match the descriptions in the paper? YesAdditional CommentsAre the data and metadata consistent with relevant minimum information or reporting standards? See GigaDB checklists for examples <a href="http://gigadb.org/site/guide" target="_blank">http://gigadb.org/site/guide</a>YesAdditional CommentsIs the data acquisition clear, complete and methodologically sound?YesAdditional CommentsIs there sufficient detail in the methods and data-processing steps to allow reproduction?YesAdditional CommentsIs there sufficient data validation and statistical analyses of data quality? YesAdditional CommentsIs the validation suitable for this type of data?YesAdditional CommentsIs there sufficient information for others to reuse this dataset or integrate it with other data?YesAdditional CommentsAny Additional Overall Comments to the AuthorThis is a review for manuscript “The first complete mitochondrial genome of Diadema antillarum (Diadematoida, Diadematidae) Majeske et al. DRR-202205-01

This is a very interesting topic. The methods and results are clearly explained. The original figures are very good and descriptive. The authors have competently analyzed the data and written a succinct manuscript. 

Marine biologists understand the legacy and impact of the Diadema epidemic from the 1980s. Therefore, it is important to help bring this species back to from the brink, if not dominance, in the Caribbean again. This could possibly happen with more systematic and molecular genomic characterizations such as this study.

Was this project part of larger project to sequence the whole Diadema genome? If so, the authors could state this and not be penalized. Due to the large number of mtDNA molecules, assembling the mitochondrial genome is commonly done in whole genome projects. Having the mtDNA properly assembled is now a great asset for conservation and population genetics. 
RecommendationAccept

---

## [Reviewer Report]

Upload additional filesDRR-202205-01/form/Review_GigaByte.docxReviewer name and names of any other individual's who aided in reviewer Remi KetchumDo you understand and agree to our policy of having open and named reviews, and having your review included with the published papers. (If no, please inform the editor that you cannot review this manuscript.)YesIs the language of sufficient quality?YesPlease add additional comments on language quality to clarify if needed
Are all data available and do they match the descriptions in the paper? YesAdditional CommentsThe GitHub is up to date but I cannot yet access the NCBI databases although numbers are provided (likely submitted but not publicly available).Are the data and metadata consistent with relevant minimum information or reporting standards? See GigaDB checklists for examples <a href="http://gigadb.org/site/guide" target="_blank">http://gigadb.org/site/guide</a>YesAdditional CommentsIs the data acquisition clear, complete and methodologically sound?YesAdditional CommentsIs there sufficient detail in the methods and data-processing steps to allow reproduction?YesAdditional CommentsI would suggest that the authors also make their alignments available to the public.Is there sufficient data validation and statistical analyses of data quality? YesAdditional CommentsIs the validation suitable for this type of data?YesAdditional CommentsIs there sufficient information for others to reuse this dataset or integrate it with other data?YesAdditional CommentsAny Additional Overall Comments to the AuthorPlease see attached docRecommendationMinor Revision

---

## [Reviewer Report]

Upload additional filesDRR-202205-01/form/gx-DR-1653683854_AK.pdfReviewer name and names of any other individual's who aided in reviewer Andreas KrohDo you understand and agree to our policy of having open and named reviews, and having your review included with the published papers. (If no, please inform the editor that you cannot review this manuscript.)YesIs the language of sufficient quality?YesPlease add additional comments on language quality to clarify if needed
Are all data available and do they match the descriptions in the paper? NoAdditional CommentsThe data was not provided together with the manuscript, so I am unable to check this. The manuscript, however, states that the data will be deposited in GenBankAre the data and metadata consistent with relevant minimum information or reporting standards? See GigaDB checklists for examples <a href="http://gigadb.org/site/guide" target="_blank">http://gigadb.org/site/guide</a>NoAdditional CommentsLovality details missing, Voucher specimen number missing, Repository institution for voucher specimen not identifiedIs the data acquisition clear, complete and methodologically sound?NoAdditional Commentssee details belowIs there sufficient detail in the methods and data-processing steps to allow reproduction?NoAdditional Commentssee details belowIs there sufficient data validation and statistical analyses of data quality? NoAdditional Commentsunclear - some detail is missing in the methods section to allow judegement - see details belowIs the validation suitable for this type of data?YesAdditional CommentsIs there sufficient information for others to reuse this dataset or integrate it with other data?NoAdditional Commentssee above and below - voucher specimen number is missing, some methodological information is missing, references to original papers providing sequences used in the analysis are missing, etc. - see details belowAny Additional Overall Comments to the AuthorThe manuscript by Majeske et al. on the mitogenome of Diadema antillarum is an interesting contribution to the phylogeny of Echinoidea. There are, however, a number of issues which should be addressed in a revised version, in my opinion.
1) Please provide coordinates for the sampling site (and a locality name) instead of a general region
2) Please provide the repository number and institution where the voucher specimen has been deposited
3) Did you verify the identification and made sure that this is D. antillarum rather than D. africanum (which allegedly has repopulated some D. antillarum habitats in the Caribbbean and GoM) – for a morphological comparison see: Rodríguez, A., Hernández, J. C., Clemente, S. & Coppard, S. E. 2013. A new species of Diadema (Echinodermata: Echinoidea: Diadematidae) from the eastern Atlantic Ocean and a neotype designation of Diadema antillarum (Philippi, 1845). Zootaxa 3636, 144-170.
4) Please report the insert size that has been targeted during library prep. (typically either 350 bp or 550 bp for the kit mentioned)
5) Explain why the S. purpuratus mitogenome was uses to map the reads rather than one of the diadematid mitogenomes
6) Please explain why the custom assembly pipeline was used rather than one of the well-established assemblers like SPAdes, Abyss, Velvet, etc.
7) Please provide a coverage graph
8) Position of the non-coding region is given in # bp – but without information which feature is considered as zero in a linearized version of the circular sequence the position is useless
9) Please explain what exactly was used for the analysis – the full nucleotid sequence including non-coding regions, just the CDS of the protein coding genes, or …?
10) Please add the reference to original papers that published the sequences you use in the tree
11) Please explain the choice of the model used in the analysis – was some Modeltest run?
12) Please provide the fasta file together with a revised version to allow checking the quality of the annotation etc.
13) Fig. 1: please provide some information on the photo shown – is this the specimen that was sampled, add this info and the locality in the caption
14) Fig. 2: add the accession numbers in the tree and highlight the new sequence
15) Please see additional minor comments in the annotated version, which is attached
Summing up, I recommend acceptance after major revision.
Kind regards
Andreas Kroh, NHM Vienna, 10/7/2022
RecommendationMajor Revision